# Prevalence of overweight and obesity and their impact on academic performance and psychological well-being among university students in 2024 in Bangladesh

Hasibul Hassain Emon[1,2], Soham Sarker[1,2], Mst. Shamima Akter Lima[2,3], Farzana Afroze Tasnim[2,4], Abdullah Al Nabil[2,4], Muhammad Ibrahim Azam[2,4], Md. Moyazzem Hossain[1,2]*

1 Department of Statistics and Data Science, Jahangirnagar University, Savar, Dhaka, Bangladesh,
2 Jahangirnagar University Research Society, Jahangirnagar University, Savar, Dhaka, Bangladesh,
3 Department of Botany, Jahangirnagar University, Savar, Dhaka, Bangladesh, 4 Department of Pharmacy, Jahangirnagar University, Savar, Dhaka, Bangladesh

* hossainmm@juniv.edu

## Abstract

### Background

The prevalence and impact of obesity and overweight is one of the main concerns among researchers worldwide. There are several consequences of overweight and obesity among students in developing countries like Bangladesh. Therefore, this study aims to explore the prevalence and impact of overweight or obesity on individuals' self-esteem and academic performance among university students in Bangladesh.

### Methods and materials

This study is based on primary data that was collected from 600 students of Jahangirnagar University using a well-structured questionnaire. The questionnaire included demographic information, the Rosenberg Self-Esteem Scale (RSES), and the International Physical Activity Questionnaire (IPAQ-7). The summary statistics, the Fisher exact test, the Chi-square test, and the Pearson correlation matrix were computed. A logistic regression model assessed the relationship between low self-esteem and other variables such as demographic characteristics, BMI status, and physical activities.

### Results

The findings indicate that 38.2% of the students are overweight or obese, and the prevalence rate is high among males. The lowest academic performance across all disciplines is observed among obese students. A negative correlation is observed between BMI and self-esteem ($r = -0.41$) and academic performance is negatively linked with overweight or obese ($r = -0.26$) Results depict that overweight/obese students are 7.11 times more likely to have low self-esteem (AOR: 7.11, 95% CI: 4.59–11.02) compared to normal students.

**Data Availability Statement:** The data is attached as a supplementary file of this manuscript.

**Funding:** The author(s) received no specific funding for this work.

**Competing interests:** The authors have declared that no competing interests exist.

## Conclusions

Overweight and obese students are more prone to lower academic performance and low self-esteem, considering physical inactivity and irregular dietary patterns. Therefore, university authorities ought to encourage students to engage in physical activities and should create awareness among students about the beneficial effects of nutritious foods and the importance of physical exercise.

## Introduction

Obesity is defined by disproportionate or unorthodox fat assemblage posing health threats and is calculated using body mass index (BMI). In accordance with the World Health Organization (WHO), contemporary global estimations demonstrate that over 39% of adults (39% of men and 40% of women) are evaluated as overweight, whereas 13% of adults (15% of women and 11% of men) are viewed as obese [1, 2]. Conversely, overweight is categorized as containing a body mass index (BMI) between 25 and 29.9 kg/m$^2$, an anthropometric measure used to evaluate excess body weight concerning height, implying conceivable health stakes [3]. A mentionworthy weight boost happens when an individual gains access to university life when young adults can determine their behavior and lifestyle alongside their recently altered food habits. Globally, a growing widespread presence of overweight and obesity is witnessed in not only developed countries but also developing countries. Researchers highlighted an increasing prevalence rate of obesity and overweight among university students in developing countries such as: in African countries (South Africa: 10.8%–24% [4]; meanwhile in Asian countries (Bangladesh: 20.8%; China: 2.9%–14.3% [5]; Malaysia: 20%–30.1%, Thailand: 31% [6], Pakistan: 13%–52.6%, and in India: 11%–37.5% [7], and eventually, the Middle and Near East (Kuwait: 42% [8], Iran 12.4%, and Turkey: 10%–47.4% [9]) implies that the undergraduate years are appealingly compelling in transforming adult behaviors, notably concerning foot habits, physical training, and further lifestyle practices [10].

Unlike children affected by parental practices and more aged grown-ups with designated and specified routines, undergraduate or graduate students encounter unusual lifestyle transitions and stressors [11]. Previous studies pointed out that university students often undergo more significant weight gain compared to their non-university-going peers [12]. Under the circumstances of this threatening prevalence rate, it is obligatory to analyze the rationales behind the adverse effects of overweight and obesity on university students' academic performance and psychological well-being. Concurrent studies indicate that non-western countries, specifically South Asian countries, seem more prone to eating disorders [13], where the following prevalences of eating disorders have been encountered by scrutinizing the eating patterns of university students. The country-wise rates are China (3.2%–9.9% females and 1.2%-2% males) [14], India (female nursing students, 4%), Malaysia Sarawak (13.7% females, 5.6% males) [15], Pakistan (17%–22.75%) [16], and Turkey (7.9%) [17].

Lifestyles with inactivity and excessive calorie consumption are factors that foster overweight and obesity. Factors that increase the likelihood of young adults becoming less physically active [18]. Diabetes, cardiovascular disorders, and some types of cancer are the physical fitness consequences of being obese [19], whereas low self-esteem beholds the role of the most vital and common mental disease [20]. Obesity is one of the most critical problems for both physical and mental health, which is directly correlated to low self-esteem. In addition to that, anxiety, stress, and depression are the most crucial mental health problems [21]. People with

these indicators face hurdles in their cognitive, emotional, and communication issues, which result in underemployment and diminished mental strength [22]. Several studies indicate that some domains of self-esteem may be more negatively affected by overweight/obese status than others. Low self-esteem, insomnia, disproportionate drowsiness, and incapability to concentrate on something are the predominant symptoms of depression [23]. At the moment of depression or low self-esteem, an individual's judgments and behavior are affected [24]. Students who have depression embrace low academic performance and life quality [25]. Recently, adverse mental health has become a global phenomenon among university students [26].

However, there is a dearth in the existing literature that focuses on the association between being obese or overweight and academic performance alongside low self-esteem among university students in the context of Bangladesh. Thus, it is necessary to explore the relationship between obesity and its impact on academic performance and low self-esteem among university students. To fill up the backdrop, this study aims to investigate the prevalence of obesity and overweight and find its impact on academic performance as well as self-esteem.

## Methods and materials

### Study settings

The primary data has been collected from students having normal weight and those who were obese or overweight at Jahangirnagar University, Bangladesh. The raw data is attached as a supplementary file (S1 File).

### Participants

All students (more than 17 thousand) are the target population in this study. The minimum sample size was calculated using $n = \frac{Z^2 p(1-p)}{d^2} \approx 384$, for $p = 0.5$, $Z = 1.96$, and $d = 0.05$ [27–29]. However, for more accuracy, a sample consisted of 600 students, and among them, 229 students were classified as obese and overweight. The inclusion criteria of the participants were valid students of Jahangirnagar University, however, students in the first year were excluded from this study.

### Study design

A stratified random sampling approach was employed to select students. Initially, the authors considered Arts and Humanities, Business Studies, Mathematical, Physical, and Biological Science, and Social Science faculty. From each faculty, students were selected from Second, Third, Fourth, and Masters levels.

### Design questionnaire and data collection

The primary data were collected using a well-structured questionnaire administered by the authors. The questionnaire consisted of sections focusing on demographic factors, physical activity features, dietary habits, academic performance, and self-esteem. To enhance clarity and dependability, the authors administered a pretest questionnaire to a small group of undergraduate students at Jahangirnagar University, Bangladesh. Furthermore, the respondents' BMI was measured using an anthropometric assessment [30]. A well-trained team administered the data-collection procedure from 25 June 2024 to 14 July 2024 under the supervision of the authors. The questionnaire used in this study for the primary data is attached as a supplementary file (S2 File).

## Variables

Several demographic information of the respondents such as age, gender, family income, family status, faculty, year of study, and residential status have been collected. The World Health Organization (WHO) guidelines of BMI classification have been followed to classify the participants as underweight (BMI<18.5 kg/m2), normal weight (BMI 18.5–24.9 kg/m2), overweight (BMI 25–29.9 kg/m2), and obese (BMI≥ 30 kg/m2) [31]. The student's academic performance was determined using their Cumulative Grade Point Average (CGPA) on a scale of 4.0. The classification of CGPA or academic achievement was segmented into three categories: high (4.00–3.51), average (3.50–3.00), and low (<3.00). The dietary habits were assessed by using inquiries addressing daily intake of red meat, fat, and fiber, as well as the percentage of intake of fruits and vegetables each day, in addition to breakfast consumption. Moreover, implementing the International Physical Activity Questionnaire (IPAQ-7), physical activity consists of variables such as the frequency of weightlifting per week, the number of hours of physical activity per week, and the amount of time spent cycling or walking per day [32]. Self-esteem or psychological well-being was evaluated using the Rosenberg Self-Esteem Scale (RSES) [33]. It consists of a scale that fluctuates from 0 to 30. In this scoring system, a greater score denotes greater self-esteem, while a lower score indicates the reverse. The Cronbach's Alpha for the IPAQ-7 and RSES are 0.765 and 0.693 respectively.

## Statistical analysis

The summary statistics of the participants are presented by the percentage distribution of each demographic characteristic, as well as summarizing the eating habits, academic achievement, and self-esteem. The association between obesity or overweight and academic performance or low self-esteem has been demonstrated via the Fisher exact test or Chi-square test. To quantify the strength of the association between obesity or overweight, academic achievement, and self-esteem, the Pearson correlation coefficient is employed, and a correlation matrix has been created for visualization. A logistic regression model was performed, incorporating characteristics such as behaviors associated with health risks, physical activity levels, dietary habits, faculty, study year, housing, family income, gender, and age. The adjusted odds ratio (AOR) and 95% confidence intervals (CIs) were measured for significant predictors. A *p*-value of less than 0.05 is considered as significant. The analysis of data has been performed using the R programming language version 4.4.1.

**Ethical approval.** The ethical approval for this study and the procedures were obtained from the Ethics Committee of Jahangirnagar University, Bangladesh with the approval reference number BBEC,JU/M 2024/05 (104) on 15 May 2024. Moreover, participants were asked to be involved in this study before starting the survey and the study objectives were informed to the participants. Respondents were adequately informed that their information would be confidential and no identifiable information would be disclosed. Furthermore, it is assumed that the participants can withdraw from the survey at any point. The consent statement was presented in the questionnaire prior to the start of the main questions of the study. The survey was started if the participants agreed to participate in the study. The Ethics Committee also reviewed the proposal and questionnaire before providing ethical approval.

## Results

The prevalence of the different weight statuses of students who participated in this study by their demographic characteristics is presented in Table 1.

Among the respondents, 60.0% of the students were more than 22 years old and 45.5% and 31.3% met the criteria for overweight or obesity among male (48.3%) and female (51.7%)

**Table 1. Prevalence of overweight and obesity by demographic characteristics (*n* = 600).**

| Variables | Labels | *n* (%) | Weight Status, n (%) | | | *p*-value of $\chi^2$ and Fisher exact test |
|---|---|---|---|---|---|---|
| | | | Under Weight, *n* (%) | Normal weight, *n* (%) | Overweight and Obese, *n* (%) | |
| **Age (Years)** | ≤22 years old | 240 (40.0) | 21 (8.8) | 137 (57.1) | 82 (34.2) | 0.245 |
| | >22 years old | 360 (60.0) | 26 (7.3) | 187 (51.9) | 147 (40.8) | |
| **Gender** | Male | 290 (48.3) | 14 (4.8) | 144 (49.7) | 132 (45.5) | <0.001 |
| | Female | 310 (51.7) | 33 (10.6) | 180 (58.1) | 97 (31.3) | |
| **Family income** | Low income | 37 (6.2) | 3 (8.1) | 22 (59.5) | 12 (32.4) | 0.075[a] |
| | Lower middle income | 294 (49.0) | 20 (6.9) | 165 (56.1) | 109 (37.0) | |
| | Higher middle income | 244 (40.7) | 23 (9.4) | 130 (53.3) | 91 (37.3) | |
| | High income | 25 (4.2) | 1 (4.0) | 7 (28.0) | 17 (68.0) | |
| **Residence** | On-campus | 442 (73.7) | 38 (8.6) | 255 (57.7) | 149 (33.7) | 0.005[a] |
| | Off-campus | 29 (4.8) | 2 (6.9) | 13 (44.8) | 14 (48.3) | |
| | Home | 129 (21.5) | 7 (5.5) | 55 (42.6) | 67 (51.9) | |
| **Faculty** | Arts and Humanities | 146 (24.3) | 14 (9.6) | 100 (68.5) | 32 (21.9) | <0.001 |
| | Business Studies | 82 (13.7) | 3 (3.7) | 26 (31.7) | 53 (64.6) | |
| | Mathematical, Physical, and Biological Science | 246 (41.0) | 18 (7.3) | 127 (51.6) | 101 (41.1) | |
| | Social Science | 126 (21.0) | 12 (9.5) | 71 (56.4) | 43 (34.1) | |
| **Study year** | Second | 163 (27.2) | 14 (8.6) | 87 (53.4) | 62 (38.0) | 0.175 |
| | Third | 201 (33.5) | 16 (8.0) | 123 (61.2) | 62 (30.8) | |
| | Fourth | 151 (25.2) | 11 (7.3) | 72 (47.7) | 68 (45.0) | |
| | Masters | 85 (14.1) | 6 (7.1) | 42 (49.4) | 37 (43.5) | |
| **Family history of obesity** | Absent | 432 (72.0) | 39 (9.0) | 256 (59.3) | 137 (31.7) | <0.001 |
| | Present | 168 (28.0) | 8 (4.8) | 68 (40.5) | 92 (54.8) | |

Note

[a]*p*-value obtained by performing the Fisher exact test

students respectively. Results show that 68.0% of students are overweight or obese and they come from higher-income families. The prevalence of students living on-campus, off-campus, and at home are 73.7%, 4.8%, and 21.5%, respectively. It is observed that 51.9% of students who live at home fall into the overweight or obese category. The majority of students live on campus and among them, 33.7% belong to the overweight and obese category. Moreover, 28.0% of all participants had a family history of obesity, and among them, 54.8% were over-weight or obese [Table 1].

**Table 2. Dietary habits and health risk behaviors of overweight and obese students (*n* = 229).**

| Variable | Labels | All | Male | Female | *p*-value |
|---|---|---|---|---|---|
| | | N (%) or M (SD) | N (%) or M (SD) | N (%) or M (SD) | |
| **Dietary variables** | | | | | |
| Eats red meat at least once a day | Yes | 48 (21.0) | 28 (58.3) | 20 (41.7) | 0.264 |
| | No | 181 (79.0) | 104 (57.5) | 77 (42.5) | |
| Try to eat fiber | Yes | 155 (67.7) | 85 (54.8) | 70 (45.2) | 0.271 |
| | No | 74 (32.3) | 47 (63.5) | 27 (36.5) | |
| Avoids fat and cholesterol | Yes | 64 (28.0) | 37 (57.8) | 27 (42.2) | 0.297 |
| | No | 165 (72.0) | 95(57.6) | 70 (42.4) | |
| Fruit and vegetables (<5 times/day) | Yes | 155 (67.7) | 81 (52.3) | 74 (47.7) | 0.024 |
| | No | 74 (32.3) | 51 (68.9) | 23 (31.1) | |
| Skipping breakfast | Yes | 88 (38.4) | 57 (64.8) | 31 (35.2) | 0.112 |
| | No | 141 (61.6) | 75 (53.2) | 66 (46.8) | |
| Number of meals a day | | 3.0 (0.8) | 3.0 (0.8) | 3.0 (0.8) | |
| Number of in-between snacks | | 1.6 (0.9) | 1.7 (0.9) | 1.6 (0.9) | |
| **Health risk behaviors** | | | | | |
| Physical activity | Low | 96 (42.0) | 45 (46.9) | 51 (53.1) | 0.005 |
| | Moderate | 107 (46.7) | 69 (61.2) | 38 (38.8) | |
| | High | 35 (15.3) | 27 (77.1) | 8 (22.9) | |
| Current tobacco use | Yes | 48 (21.0) | 38 (79.2) | 10 (20.8) | 0.001 |
| | No | 181 (79.0) | 94 (51.9) | 87 (48.1) | |
| Binge drinking (past month) | Yes | 26 (11.4) | 22 (84.6) | 4 (15.4) | 0.006 |
| | No | 203 (88.6) | 110 (54.2) | 93 (45.8) | |

Table 2 presents the dietary habits and health risk behaviors of 229 overweight and obese students by sex of the respondent. Among them, 58.3% of males and 41.7% of females consumed red meat daily, with 21% of all students consuming red meat daily. Additionally, 54.8% of males and 45.2% of females tried to eat fiber, while only 28% of overweight and obese students avoided fat and cholesterol-containing foods. Furthermore, 67.7% of students consumed fruits and vegetables at least 5 times daily. Skipping breakfast was more common among males (64.8%) than females (35.2%). Almost half (46.7%) of the overweight and obese students engaged in moderate physical activity, whereas only 15.3% reported they were involved in high physical activity. Findings suggested that 21% of these students, the majority males (79.2%) reported that they consume tobacco. It is observed that 11.4% of respondents reported binge drinking in the previous month, with male students (84.6%) reporting this behavior much more frequently than female students (15.4%).

Table 3 illustrates students' academic performance according to their BMI status and faculties. Students who maintained a normal weight consistently had the highest average cumulative grade point average (CGPA), ranging from 3.57 in Business Studies to 3.38 in Arts and Humanities faculties. Underweight students also demonstrated strong performance, achieving average CGPAs ranging from 3.40 in Arts and Humanities to 3.60 in Business Studies faculties. Nevertheless, students classified as overweight or obese across all academic disciplines exhibited the lowest mean CGPAs, ranging from 3.24 in the Social Sciences to 3.32 in the Business Studies faculties.

Students with a normal weight had a larger percentage of high academic achievement (73.8%) compared to underweight (10.3%) and overweight/obese students (15.9%). Moreover, students who were overweight or obese had a significantly higher percentage (58.8%) of lower

**Table 3. Academic performance of students according to BMI status and faculties (_n_ = 600).**

| Faculty | BMI Status | CGPA (mean ± SD) |
|---|---|---|
| Arts and Humanities | Normal weight | 3.38 ± 0.24 |
|  | Overweight and Obese | 3.26 ± 0.26 |
|  | Underweight | 3.40 ± 0.13 |
| Business Studies | Normal weight | 3.57 ± 0.23 |
|  | Overweight and Obese | 3.32 ± 0.24 |
|  | Underweight | 3.60 ± 0.28 |
| Mathematical, Physical, and Biological | Normal weight | 3.53 ± 0.26 |
|  | Overweight and Obese | 3.28 ± 0.30 |
|  | Underweight | 3.49 ± 0.30 |
| Social Science | Normal weight | 3.50 ± 0.29 |
|  | Overweight and Obese | 3.24 ± 0.27 |
|  | Underweight | 3.58 ± 0.21 |

academic performance compared to underweight students (5.0%) and students with a normal weight (36.2%) [Fig 1].

The correlation between BMI status and self-esteem status among the 600 students is illustrated in Fig 2(A). It revealed that 42.0% of students with a normal weight had high self-esteem. Among the students, 25% of those classified as overweight or obese reported experiencing low self-esteem. The associations among self-esteem, academic achievement, and overweight/obesity are depicted in Fig 2(B). The results revealed a significant inverse relationship between obesity/overweight with self-esteem ($r$ = -0.41) and academic performance ($r$ = -0.26). A strong correlation is observed between an increase in overweight and obesity and a decrease in academic performance and self-esteem.

The unadjusted model showed that students aged more than 22 years had 1.47 times higher odds (COR: 1.47, 95% CI: 1.05–2.07) of experiencing low self-esteem compared to those who were 22 years old or younger. In addition, the unadjusted model showed that female students had a 30.0% lower likelihood of having low self-esteem than male students (COR: 0.70, 95% CI: 0.50–0.98). Students who were from higher-income families showed a 2.46-fold higher likelihood (COR: 2.46, 95% CI: 0.87–6.77) of reporting low self-esteem compared to those from lower-income families. Findings indicate that students in the Business Studies faculty had a 1.93 times higher chance of developing low self-esteem than students of the Arts and Humanities faculty (COR: 1.93, 95% CI: 1.10–3.38). Similarly, students in the Social Science faculty had a 1.91 times higher chance of developing low self-esteem than students in the Arts and Humanities faculty. Students with a family history of obesity were 2.10 times more likely to have low self-esteem compared to those without such a history (COR: 2.10, 95% CI: 1.46–3.02). Students who performed at a medium academic level had a 48.0% reduced probability of experiencing low self-esteem (COR: 0.52, 95% CI: 0.36–0.76). On the other hand, students who performed at a high academic level had a 70.0% reduced chance of experiencing low self-esteem than students who performed at a low academic performance (COR: 0.30, 95% CI: 0.18–0.51). Similarly, individuals with overweight and obese BMI status had a 7.29 times greater likelihood of having low self-esteem (COR: 7.29, 95% CI: 4.97–10.68) compared to those with normal weight [Table 4].

Nevertheless, the adjusted model revealed that age, faculty, family history of obesity, and BMI status were identified as significant factors of low self-esteem (p<0.05). Students with a family history of obesity had a 1.59 times greater likelihood of having low self-esteem (AOR: 1.59, 95% CI: 1.03–2.44) compared to students without a family history of obesity, despite

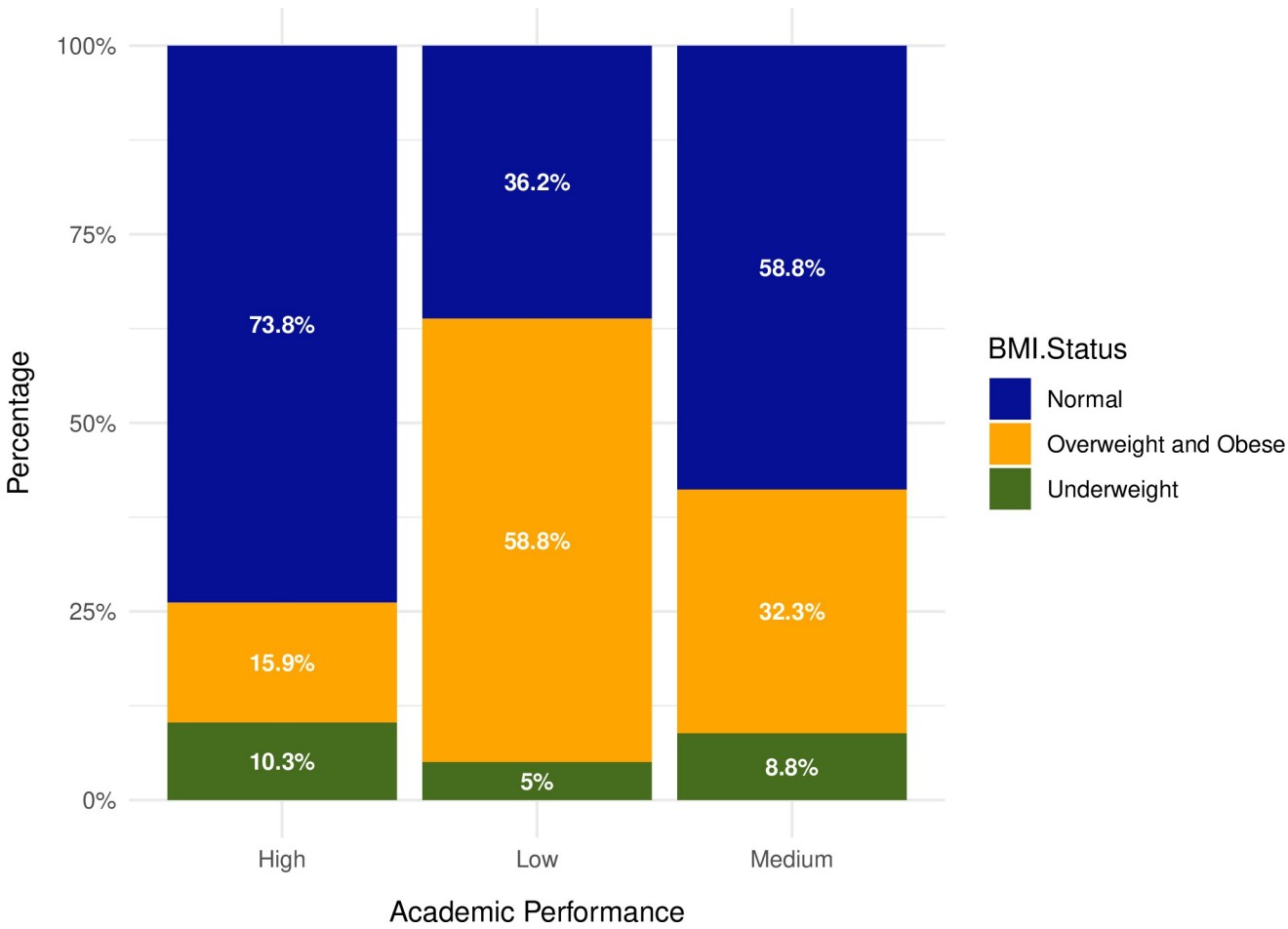

**Fig 1. Percentage of students by BMI status and their academic performance.**

accounting for the influence of other confounding variables. The study also found that overweight and obese students had a 7.11 times higher likelihood of experiencing low self-esteem compared to students of normal weight (AOR: 7.11, 95% CI: 4.59–11.02) after accounting for all relevant factors in the logistic regression analysis [Table 4].

## Discussion

This study examined the prevalence of overweight and obesity, and how they affect self-esteem and academic achievement. The authors also examined the eating habits and health-risky behaviors among university students. Among the students who participated in this study, 54.8% of students were overweight or obese, and 28.0% of students had a family history of obesity. More than half of the students (51.9%) who lived at home were categorized as overweight or obese. Findings revealed that overweight and obesity are more prevalent among male students (45.5%) compared to female students (31.3%) which is consistent with another study. The results of this study showed an association between university students' academic achievement and their BMI. Normal-weight students typically achieved higher average CGPAs than overweight or obese students across all faculties. Previous studies also pointed out this gender-wise variation among university students [26]. The reason behind these differences may be that female students are more aware of their body weight than males and maintain their

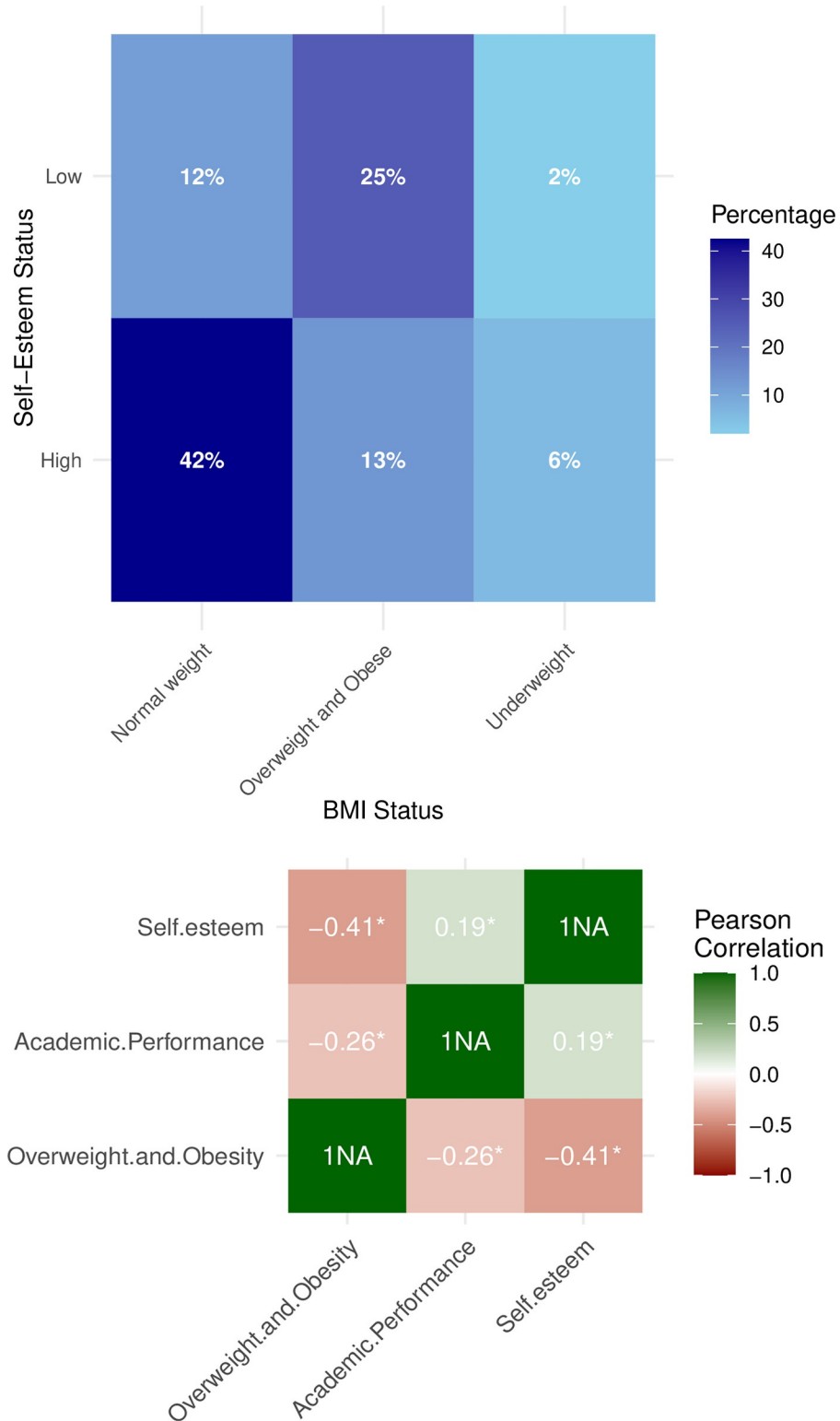

**Fig 2.** (a) Percent distribution of BMI status vs self-esteem status ($n = 600$), (b) Correlation matrix between overweight and obesity, self-esteem and academic performance ($n = 229$). *: Significant at p < 0.05.

**Table 4. Results of the logistic regression model (_n_ = 600).**

| Variables | Labels | Unadjusted Model | | Adjusted Model | |
|---|---|---|---|---|---|
| | | COR | 95% CI | AOR | 95% CI |
| Age (Years) | ≤22 years old | Ref. | - | Ref. | - |
| | >22 years old | 1.47** | 1.05–2.07 | 1.70* | 0.97–2.98 |
| Gender | Male | Ref. | - | Ref. | - |
| | Female | 0.70** | 0.50–0.98 | 0.92 | 0.61–1.40 |
| Family income | Low income | Ref. | - | Ref. | - |
| | Lower middle income | 1.07 | 0.52–2.16 | 0.93 | 0.41–2.12 |
| | Higher middle income | 0.92 | 0.45–1.89 | 0.79 | 0.34–1.84 |
| | High income | 2.46* | 0.87–6.97 | 1.45 | 0.42–4.97 |
| Residence | On-campus | Ref. | - | Ref. | - |
| | Off-campus | 1.15 | 0.53–2.47 | 0.90 | 0.36–2.25 |
| | Home | 1.13 | 0.76–1.69 | 0.78 | 0.47–1.29 |
| Faculty | Arts and Humanities | Ref. | - | Ref. | - |
| | Business Studies | 1.93** | 1.10–3.38 | 0.75 | 0.37–1.53 |
| | Mathematical, Physical, and Biological Science | 1.34 | 0.86–2.07 | 0.91 | 0.54–1.55 |
| | Social Science | 1.91** | 1.16–3.14 | 1.81** | 1.01–3.22 |
| Study year | Second | Ref. | - | Ref. | - |
| | Third | 1.00 | 0.65–1.53 | 0.87 | 0.48–1.55 |
| | Fourth | 1.33 | 0.84–2.09 | 0.77 | 0.38–1.56 |
| | Masters | 1.08 | 0.63–1.86 | 0.64 | 0.29–1.41 |
| Family history of obesity | Absent | Ref. | - | Ref. | - |
| | Present | 2.10*** | 1.46–3.02 | 1.59** | 1.03–2.44 |
| Academic performance | Low | Ref. | - | Ref. | - |
| | Medium | 0.52*** | 0.36–0.76 | 0.82 | 0.53–1.27 |
| | High | 0.30*** | 0.18–0.51 | 0.66 | 0.36–1.22 |
| BMI status | Normal weight | Ref. | - | Ref. | - |
| | Underweight | 1.26 | 0.62–2.57 | 1.29 | 0.62–2.66 |
| | Overweight and Obese | 7.29*** | 4.97–10.68 | 7.11*** | 4.59–11.02 |
| Physical activity | Low | Ref. | - | Ref. | - |
| | Moderate | 0.91 | 0.64–1.30 | 0.73 | 0.48–1.11 |
| | High | 1.10 | 0.66–1.83 | 0.90 | 0.49–1.65 |

Note:

*: $0.05 \leq p < 0.1$

**: $p < 0.05$

***: $p < 0.001$; Target variable (Low = 1, High = 0)

nutritious diet. Researchers also highlighted the same reason for the difference: women typically prefer more nutritious food options and display a greater worry regarding weight management than men [15].

It is observed that most students residing in university residential halls were found to have a normal weight. A previous study also found similar results [34]. Nevertheless, a previous study pointed out that students who resided off-campus had a higher tendency to be overweight or obese compared to students who lived on campus or with their parents [35]. Students who lived at home were more likely to be obese because of more food choices and may visit nearby restaurants frequently to consume fast food. A study showed that eating more fast food led to obesity [36]. Conversely, limited food choices are available in the university area,

and some students spend a limited budget on food. A study among adults in Edmonton, Canada also found the same result [37].

Moreover, among all participants, 28.0% of students had a family history of obesity, and 54.8% of students were obese or overweight. This prevalence is higher compared to the results of previous studies [38, 39]. In addition to typical familial lifestyle habits and eating patterns that promote weight gain, genetic factors that affect metabolism and fat storage can be responsible for the higher prevalence of obesity among those with a family history of obesity. Additionally, there was a significant positive relationship found between the number of obese family members and being overweight or obese [40]. Smoking and consuming alcohol constituted a significant risk factor for health among students. One possible explanation for the increased tobacco usage among students could be the tight schedule of the year-end exams, which involve several exams, class evaluations, homework, and laboratory work, creating a stressful and demanding environment. Another study's findings supported this study's findings and researchers mentioned that excessive episodic drinking increased the chance of being overweight [41]. In previous studies, researchers observed that exam stress significantly affects the dietary habits of students [42].

The findings of this study revealed the correlation between BMI status and the academic performance of university students. Across all faculties, students with a normal weight consistently attained higher average CGPAs than overweight or obese students. Other studies also highlighted that higher weight lowers educational achievement among children and young individuals [43]. It is observed that obesity has a detrimental impact on concentration and academic performance. Researchers highlighted an association between sleep apnea, weariness, and poor academic performance among students [44, 45]. Moreover, overweight and obesity not only affect academic achievement but also have an impact on self-esteem. Results revealed that overweight or obese students had 7.29 times higher odds of experiencing low self-esteem compared to students with a normal weight. This result is consistent with the findings of previous research [46]. Overweight or obese may have reduced self-esteem among university students because of societal and cultural pressures, bullying and taunting, absorbed weight stigma, and social isolation. A study pointed out that teasing related to body weight and physical appearance may lessen self-esteem [47, 48].

It is observed that students who have poor academic performance are more likely to experience low self-esteem which is consistent with another study [49, 50]. The student's academic achievement may be impacted by their overweight or obese status. Several studies showed that increased body weight have a negative effect on academic performance [51, 52]. Moreover, results demonstrated that students who engage in high levels of physical exercise have 10% less odds of experiencing low self-esteem compared to those with low levels of physical activity. Researchers pointed out that engaging in physical activities directly enhances self-esteem [53]. This may be explained by the fact that exercise causes endorphins to be secreted, which improve mood and lower stress levels while also greatly increasing self-esteem [54].

## Limitations of the study

This study has a few limitations. This study was conducted using a cross-sectional approach, the causal inference is not suitable here. The data about self-esteem, weekly physical activity, and food habits may be subject to memory bias or social desirability bias. Moreover, the conclusions or outcomes of this study may not be indicative of the entire population and may be different from other samples collected from other universities, or various age groups. This study is limited to a single university due to a lack of funding support; nevertheless, the authors intend to undertake a nationwide survey soon. Finally, there may be other factors not included

in this study that influence the prevalence of overweight and obesity among university students, as well as their influence on academic performance and psychological well-being.

## Conclusions

Male students who live in their own homes have higher rates of overweight and obesity, which can be attributed to food habits and other health-risky behaviors. Students who are overweight or obese have lower academic performance and low self-esteem due to their lifestyle habits and lack of physical activities. The authors recommended that along with focusing their study, students should participate in regular physical activities and maintain healthy food habits. It is also important to design and implement appropriate strategies to promote healthier lifestyles and enhance psychological well-being among university students to improve academic performance. A further study considering a countrywide survey may be helpful to the policymakers in making a policy in this regard. A nationwide survey will be potential research for further study. The authors intend to create a Shiny App that will enable the researchers to determine how obesity and other factors affect students' academic performance and mental health.

## Supporting information

**S1 File. Raw data file.**
(XLSX)

**S2 File. Questionnaire.**
(PDF)

**S3 File. STROBE statement—checklist of items that should be included in reports of *cross-sectional studies*.**
(DOCX)

## Acknowledgments

The authors are grateful to the participants for providing the information and consent for the publication of the survey results without any identifiable information. The authors are also thankful to the academic editor and two reviewers for their insightful comments and feedback that helped to enhance the quality and readability of the manuscript.

## Author Contributions

**Conceptualization:** Hasibul Hassain Emon, Md. Moyazzem Hossain.

**Data curation:** Soham Sarker, Mst. Shamima Akter Lima, Farzana Afroze Tasnim, Abdullah Al Nabil, Muhammad Ibrahim Azam.

**Formal analysis:** Hasibul Hassain Emon.

**Methodology:** Hasibul Hassain Emon, Soham Sarker, Mst. Shamima Akter Lima, Farzana Afroze Tasnim, Md. Moyazzem Hossain.

**Supervision:** Md. Moyazzem Hossain.

**Validation:** Md. Moyazzem Hossain.

**Visualization:** Hasibul Hassain Emon.

**Writing – original draft:** Hasibul Hassain Emon, Soham Sarker, Abdullah Al Nabil, Muhammad Ibrahim Azam.

**Writing – review & editing:** Md. Moyazzem Hossain.

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
