## [Decision Letter · Decision Letter 0]

14 Oct 2024

PONE-D-24-36391Prevalence of Overweight and Obesity and their Impact on Academic Performance and Psychological Well-being among University StudentsPLOS ONE

Dear Dr. Hossain,

Thank you for submitting your manuscript to PLOS ONE. After careful consideration, we feel that it has merit but does not fully meet PLOS ONE’s publication criteria as it currently stands. Therefore, we invite you to submit a revised version of the manuscript that addresses the points raised during the review process.

Please submit your revised manuscript by Nov 28 2024 11:59PM. If you will need more time than this to complete your revisions, please reply to this message or contact the journal office at plosone@plos.org. Please include the following items when submitting your revised manuscript:A rebuttal letter that responds to each point raised by the academic editor and reviewer(s). You should upload this letter as a separate file labeled 'Response to Reviewers'.A marked-up copy of your manuscript that highlights changes made to the original version. You should upload this as a separate file labeled 'Revised Manuscript with Track Changes'.An unmarked version of your revised paper without tracked changes. You should upload this as a separate file labeled 'Manuscript'.

We look forward to receiving your revised manuscript.

Kind regards,

Sirwan Khalid Ahmed

Academic Editor

PLOS ONE

Journal Requirements:

2. In the ethics statement in the Methods, you have specified that verbal consent was obtained. Please provide additional details regarding how this consent was documented and witnessed, and state whether this was approved by the IRB

3. You indicated that you had ethical approval for your study. In your Methods section, please ensure you have also stated whether you obtained consent from parents or guardians of the minors included in the study or whether the research ethics committee or IRB specifically waived the need for their consent.

4. In this instance it seems there may be acceptable restrictions in place that prevent the public sharing of your minimal data. However, in line with our goal of ensuring long-term data availability to all interested researchers, PLOS’ Data Policy states that authors cannot be the sole named individuals responsible for ensuring data access (http://journals.plos.org/plosone/s/data-availability#loc-acceptable-data-sharing-methods).

Reviewers' comments:

Reviewer's Responses to Questions

**Comments to the Author**

1. Is the manuscript technically sound, and do the data support the conclusions?

Reviewer #1: Partly

Reviewer #2: Partly

2. Has the statistical analysis been performed appropriately and rigorously? 

Reviewer #1: Yes

Reviewer #2: No

3. Have the authors made all data underlying the findings in their manuscript fully available?

Reviewer #1: Yes

Reviewer #2: Yes

4. Is the manuscript presented in an intelligible fashion and written in standard English?

Reviewer #1: Yes

Reviewer #2: Yes

5. Review Comments to the Author

Reviewer #1: many thanks for sending this paper to review. i have reviewed and i wrote my comments separately in the different attached file. the topic is good and informative for the students and as well as for the general population. this topic is important since obesity can make different health issues inside community.

Reviewer #2: Generally, the topic is good. The introduction is sound and has alluded to the research gap. However, the objectives mentioned at the end of the introduction is only part of what the author actually report in the results and discussion. The method is too short. Need to elaborate more on the tools used including whether it is a validated questionnaire or if its newly developed, then the author need to explain how it was developed. The results are more descriptive. In addition, it is not right for the author to claim association based on univariate analysis, hence it should not be presented and drawn conclusion based on it.

6. PLOS authors have the option to publish the peer review history of their article (what does this mean?). If published, this will include your full peer review and any attached files.

Reviewer #1: **Yes: **Kochr Ali Mahmood

Reviewer #2: No

---

## [Author Response · Author response to Decision Letter 0]

31 Oct 2024

Author responses to the review comments:

We would like to express our sincere gratitude to the reviewers and the Editor and reviewers for their valuable comments. We have considered all the comments made by the reviewers and thoroughly revised and formatted the manuscript accordingly. A detailed response to each of the comments is provided below:

Author's Response to the Academic Editor:

Thank you very much for carefully checking the manuscript and providing insightful comments. 

All required files are uploaded to the journal system. Revised texts are in red color.

Author's Response to the Journal Requirements:

1. Thanks. We revised the manuscript following the PLOS ONE style. Revised texts are in red color.

Page: 1-20

2. Thanks. We add it in the Methods section. Before starting the survey, the respondents were informed about the objective of the study and asked whether they wanted to participate in the study voluntarily or not. If they aged then the survey continued. Revised texts are in red color.

Page: 6

3. Thanks. We received ethical approval from the IRB of Jahangirnagar University, Bangladesh. Additionally, we take consent from the participants because we value the privacy of the respondents as well. Revised texts are in red color.

Page: 6

4. Thanks. We revised the data availability statement as “The data is attached as a supplementary file of this manuscript.” The data is attached as a supplementary file of the revised version of the manuscript. Revised texts are in red color.

Page: 16

Author's Response to the Reviewer 1 Comments:

Thank you very much for carefully checking the manuscript and providing insightful comments. 

Revised texts are in red color.

Thanks for your suggestion. We revise the title of the manuscript. Revised texts are in red color.

Page: 1

We appreciate your comments. We revise the Abstract. 

 Revised texts are in red color.

Page: 1-2

Thanks. 

Thank you for your comments and feedback. Revised texts are in red color.

Page: 4

We are thankful to you for carefully checking the manuscript. We revised the Methods section. Revised texts are in red color.

Page: 4-6

Thanks for highlighting this point. This study is limited to a single university due to a lack of funding support; nevertheless, they intend to soon undertake a nationwide survey. We mention it in the Limitations section. Revised texts are in red color.

Page: 14

Thanks. We revised the manuscript. Revised texts are in red color.

Page: 4-6

Thanks. Actually, 600 students is the total sample size. We revised the manuscript. Revised texts are in red color.

Page: 4-6

Thanks. We add the inclusion and exclusion criteria. We now mentioned that we took oral consent from the respondents in the revised version of the manuscript. Revised texts are in red color.

Page: 4-6

Thank you. We add it in the revised manuscript. Revised texts are in red color.

Page: 6

Thanks for your inspirational comments. 

Thank you very much. We have revised the Discussion section. Revised texts are in red color.

Page: 12-14

Thank you. We revised the Conclusion section. Revised texts are in red color.

Page: 15

Thanks. We add the recommendation and direction of further study at the end of the Conclusion section. Revised texts are in red color.

Page: 15

Thanks. We revised the reference list. We add the citations using Mendeley software and follow the PLOS ONE style. Revised texts are in red color.

Page: 16-20

Author's Response to the Reviewer 2 Comments:

Thank you very much for your comments and feedback. We appreciate your comments. We revised the manuscript as per your comments and feedback. 

The questionnaire which includes background information was developed based on the existing literature and self-efficacy. However, we also used some widely used and validated scales as we mentioned in the Methods section. 

The association is identified by correlation and logistic regression model. Revised texts are in red color.

Page: 1-15

Finally, the revised manuscript has been produced following the valuable comments and suggestions of the reviewers. Once again, we would like to thank the reviewers for their sincere dedication, professional insights, and earnest cooperation in reviewing the manuscript.

---

## [Decision Letter · Decision Letter 1]

11 Nov 2024

PONE-D-24-36391R1Prevalence of Overweight and Obesity and their Impact on Academic Performance and Psychological Well-being among University Students in 2024 in BangladeshPLOS ONE

Dear Dr. Hossain, 

Thank you for submitting your manuscript to PLOS ONE. After careful consideration, we feel that it has merit but does not fully meet PLOS ONE’s publication criteria as it currently stands. Therefore, we invite you to submit a revised version of the manuscript that addresses the points raised during the review process.

Please submit your revised manuscript by Dec 26 2024 11:59PM. If you will need more time than this to complete your revisions, please reply to this message or contact the journal office at plosone@plos.org. Please include the following items when submitting your revised manuscript:A rebuttal letter that responds to each point raised by the academic editor and reviewer(s). You should upload this letter as a separate file labeled 'Response to Reviewers'.A marked-up copy of your manuscript that highlights changes made to the original version. You should upload this as a separate file labeled 'Revised Manuscript with Track Changes'.An unmarked version of your revised paper without tracked changes. You should upload this as a separate file labeled 'Manuscript'.If applicable, we recommend that you deposit your laboratory protocols in protocols.io to enhance the reproducibility of your results. Protocols.io assigns your protocol its own identifier (DOI) so that it can be cited independently in the future. For instructions see: https://journals.plos.org/plosone/s/submission-guidelines#loc-laboratory-protocols. Additionally, PLOS ONE offers an option for publishing peer-reviewed Lab Protocol articles, which describe protocols hosted on protocols.io. Read more information on sharing protocols at https://plos.org/protocols?utm_medium=editorial-email&utm_source=authorletters&utm_campaign=protocols.

We look forward to receiving your revised manuscript.

Kind regards,

Sirwan Khalid Ahmed

Academic Editor

PLOS ONE

Journal Requirements:

**Additional Editor Comments:**
Please follow the STROBE checklist, ensuring that all required subheadings are included, particularly in the Methods section. Additionally, provide this checklist as a supplementary file.Attach the finalized English version of the questionnaires.For the sample size calculation section, please include supportive references from recently published papers (2024) that discuss how to choose a sampling technique and determine sample size for research  for sample size calculation part. 

Reviewers' comments:

Reviewer's Responses to Questions

**Comments to the Author**

1. If the authors have adequately addressed your comments raised in a previous round of review and you feel that this manuscript is now acceptable for publication, you may indicate that here to bypass the “Comments to the Author” section, enter your conflict of interest statement in the “Confidential to Editor” section, and submit your "Accept" recommendation.

Reviewer #1: All comments have been addressed

2. Is the manuscript technically sound, and do the data support the conclusions?

Reviewer #1: Yes

3. Has the statistical analysis been performed appropriately and rigorously? 

Reviewer #1: Yes

4. Have the authors made all data underlying the findings in their manuscript fully available?

Reviewer #1: Yes

5. Is the manuscript presented in an intelligible fashion and written in standard English?

Reviewer #1: Yes

6. Review Comments to the Author

Reviewer #1: The authors tried to thee best, that's why it is ready to publish i think. Their method was clear, as well as the statistical tests were good.

7. PLOS authors have the option to publish the peer review history of their article (what does this mean?). If published, this will include your full peer review and any attached files.

Reviewer #1: **Yes: **Assistant Professor Dr Kochr Ali Mahmood

---

## [Author Response · Author response to Decision Letter 1]

11 Nov 2024

Author responses to the review comments:

We would like to express our sincere gratitude to the reviewers and the Editors for their valuable comments. We have considered all the comments made by the reviewers and thoroughly revised and formatted the manuscript accordingly. A detailed response to each of the comments is provided below:

Thank you very much for carefully checking the manuscript and providing insightful comments. 

All required files are uploaded to the journal system. Revised texts are in red color.

Author's Response to Journal Requirements:

Thanks. We checked the reference list and we confirm that they are correct. Revised texts are in red color.

Page: 17-21

Author's Response to Additional Editor Comments:

Thank you very much. We follow the STROBE checklist and submit it as a supplementary file. 

Thanks. We add the questionnaire and submit it as a supplementary file (S2 File). 

Thanks. We add the references published in 2024. Revised texts are in red color.

Page: 4

Finally, the revised manuscript has been produced following the valuable comments and suggestions of the reviewers. Once again, we would like to thank the reviewers for their sincere dedication, professional insights, and earnest cooperation in reviewing the manuscript.

---

## [Editor Report · Decision Letter 2]

13 Nov 2024

PONE-D-24-36391R2Prevalence of Overweight and Obesity and their Impact on Academic Performance and Psychological Well-being among University Students in 2024 in BangladeshPLOS ONE

Dear Dr. Hossain,

Thank you for submitting your manuscript to PLOS ONE. After careful consideration, we feel that it has merit but does not fully meet PLOS ONE’s publication criteria as it currently stands. Therefore, we invite you to submit a revised version of the manuscript that addresses the points raised during the review process.

**ACADEMIC EDITOR:** 1. Please remove the irrelevant  citations such as 28, 29 and 30, and avoid of self citations.2. Please add another subheading in the method section regarding Validity and Reliability including Cronbach alpha values for all scales.3. The first paragraph of discussion should be your main results4. Add recommendation for future research 

We look forward to receiving your revised manuscript.

Kind regards,

Sirwan Khalid Ahmed

Academic Editor

PLOS ONE
---

## [Author Response · Author response to Decision Letter 2]

15 Nov 2024

Author responses to the review comments:

We would like to express our sincere gratitude to the reviewers and the Editors for their valuable comments. We have considered all the comments made by the reviewers and thoroughly revised and formatted the manuscript accordingly. A detailed response to each of the comments is provided below:

Author's Response to Academic Editor Comments:

Thank you very much for carefully checking the manuscript and providing insightful comments. 

All required files are uploaded to the journal system. Revised texts are in red color.

1. Thank you again for a few additional comments. We removed the citations 28, 29, and 30. 

Revised texts are in red color.

Page: 4

2. Thanks. We add the Cronbach alpha values for IPAQ-7 and RSES. Revised texts are in red color.

Page: 6

3. Thanks. We revised the first paragraph of the Discussion section as per your comments. Revised texts are in red color.

Page: 13

4. Thanks. In the final part of the conclusion section, we add future research directions. Revised texts are in red color.

Page: 15

Author's Response to Journal Requirements:

Thanks. We checked the reference list and we confirm that they are correct. 

Page: 17-21

Finally, the revised manuscript has been produced following the valuable comments and suggestions of the reviewers. Once again, we would like to thank the reviewers for their sincere dedication, professional insights, and earnest cooperation in reviewing the manuscript.

---

## [Editor Report · Decision Letter 3]

25 Nov 2024

Prevalence of Overweight and Obesity and their Impact on Academic Performance and Psychological Well-being among University Students in 2024 in Bangladesh

PONE-D-24-36391R3

Dear Dr. Moyazzem Hossain,

We’re pleased to inform you that your manuscript has been judged scientifically suitable for publication and will be formally accepted for publication once it meets all outstanding technical requirements.

Kind regards,

Sirwan Khalid Ahmed

Academic Editor

PLOS ONE
---

## [Editor Report · Acceptance letter]

28 Nov 2024

PONE-D-24-36391R3 

PLOS ONE

Dear Dr. Hossain, 

I'm pleased to inform you that your manuscript has been deemed suitable for publication in PLOS ONE. Congratulations! Your manuscript is now being handed over to our production team.

Kind regards, 

on behalf of

Dr. Sirwan Khalid Ahmed 

Academic Editor

PLOS ONE